# Toxigenic Fungi and Co-Occurring Mycotoxins in Maize (*Zea mayz* L.) Samples from the Highlands and Coast of Ecuador

**DOI:** 10.3390/foods14152630

**Published:** 2025-07-26

**Authors:** Héctor Palacios-Cabrera, Juliana Fracari, Marina Venturini Copetti, Carlos Augusto Mallmann, Marcelo Almeida, María Raquel Meléndez-Jácome, Wilson Vásquez-Castillo

**Affiliations:** 1Escuela de Nutrición, Facultad de Ciencias de la Salud, Universidad Espíritu Santo, Samborondón 0901952, Ecuador; hpalacioscabrera@gmail.com; 2Departamento de Tecnologia e Ciência dos Alimentos (DTCA), Centro de Ciências Rurais (CCR), Universidade Federal de Santa Maria, UFSM, Santa Maria 97105-900, Brazil; jufracari@gmail.com (J.F.); marina.copetti@ufsm.br (M.V.C.); carlos.mallmann@ufsm.br (C.A.M.); 3Ingeniería Agroindustrial, Universidad de las Américas, UDLA, Quito 170513, Ecuador; marcelo.almeida.marcano@udla.edu.ec (M.A.); maria.melendez@udla.edu.ec (M.R.M.-J.)

**Keywords:** *Fusarium*, *Aspergillus*, mycotoxins, maize, Ecuador

## Abstract

Maize is a key crop in Ecuador for both human and animal consumption. Its vulnerability to fungal contamination and mycotoxins poses risks to food safety. The aim of this study was to analyze the occurrence of fungi and mycotoxins in maize grown in different regions of Ecuador (29 localities) and postharvest factors influencing contamination. Fungal identification was performed through culturing and morphological analysis. Analysis of multi-toxins was carried out using liquid chromatography coupled with mass spectrometry (LC-MS/MS). Statistical analyses included PCA and linear regression models. Fungal contamination was found in 93.3% of samples; mycotoxins were present in 90%. *Fusarium* and *Aspergillus* were dominant. Fumonisins (66.6%), zearalenone (30%), aflatoxins (16.7%), and trichothecenes B (13.3%) were the most prevalent. Co-occurrence of up to three mycotoxins per sample was observed, more frequent on the coast. Grain moisture and temperature were strongly correlated with contamination levels. The study reveals widespread contamination of Ecuadorian maize, with environmental and postharvest factors playing key roles. This poses a food safety concern, highlighting the need for improved storage and monitoring systems.

## 1. Introduction

Maize (*Zea mays* L.) has its center of origin in Mexico and has been cultivated for more than 8000 years in various parts of the world, thanks to its great capacity to adapt to different latitudes. It is the cereal with the highest world production [1]. Due to its excellent nutritional characteristics, this cereal is used for food and as an ingredient also [2,3]. Being an energetic food, it presents many applications and makes it a commercial product of great importance [4,5,6].

Maize is the most cultivated cereal in Ecuador and has great relevance for the role it plays in the food security of the population [7], as well as for the entire economy of the Andean region and Latin America (INIAP, sf). This is a crop that has managed to adapt to all altitudinal levels and a variety of soil types [8].

In Ecuador, its cultivation is divided into four regions: Littoral, Sierra, Amazonia, and Galapagos. The country has different types of climates, such as inter-Andean tropical, moderate subtropical, cold, coastal humid tropical, tropical monsoon, dry tropical, and eastern tropical. Rainfall varies between 200 and 5000 mm/year in the different regions. Relative humidity is considered seasonal and can reach up to 85% in mountainous areas or 100% in coastal and eastern regions [9]. The relationship between climatic factors and maize production is relevant; the amount of rainfall received during the crop cycle until harvest influences yields.

The Ecuadorian Sierra, which is located at an average altitude of 2300 to 2800 m above sea level, allows the cultivation of maize known as “maíz de los Andes” or “maíz suave”, which is mainly used for human consumption, flour production, or direct consumption [10,11]. In turn, “hard yellow maize” is produced in lowland or coastal regions of Ecuador, at less than 1200 m above sea level. The country reaches a planted area of about 300,000 ha [12,13] and about 80% of all this yellow hard maize produced in Ecuador is destined for feed processing [14]. In contrast, in the inter-Andean valleys, mainly “maíz morocho” is grown, which is a grain with high starch content and is soft inside. This grain is used for human and animal feed [11]. In the last 20 years, there has been a reduction in the areas cultivated with yellow hard maize, while the areas dedicated to the cultivation of soft white maize have remained constant [11].

The most widely used cultivation method in the country and considered traditional is in rows and in small extensions of land. [8] describe that there is no clear information on the microbiological quality of maize stored in silos, even though this food and its derivatives have a high potential for contamination by mycotoxins, secondary metabolites produced by various fungal species that infect the grain. The incidence of fungal growth and mycotoxin production is the result of the interaction of the fungus, the host, and the environment. The combination of these three factors results in fungal colonization on a substrate and mycotoxin production [15,16,17].

Fungal species that colonize grains are divided into two main groups: field fungi that manage to infect the grain still in the field because environmental conditions are favorable for their development; and storage fungi that infect maize shortly after harvest and during the storage period [18,19,20].

Field fungi can be phytopathogenic or commensal, and their establishment depends on climatic conditions, which makes their control difficult. *Fusarium* sp. is the main contaminant and the causal agent of ear maize deterioration. They are endemic, and the species with the highest incidence are *Fusarium graminearum*, *Fusarium verticillioides*, and *Fusarium subglutinans*. Invasion of these strains is mainly associated with insect-damaged tissues [21,22,23]. Studies also emphasize the importance of *Aspergillus flavus* as a pre-harvest infectious agent and subsequent contamination of maize by aflatoxin group mycotoxins at unacceptable levels [21,24,25,26].

Because cobs and kernels are relatively large, high moisture conditions in postharvest result in slow drying. Consequently, both field and storage fungi have the opportunity to become established at this transitional time [15,27,28].

Storage fungi tend to be present in high numbers and in all types of material, such as air, dust, grains, and seeds, where they proliferate when they find the right conditions [15]. Xerophilic *Aspergillus* species, especially *A. ruber*, *A. pseudoglaucus*, and *A. chevalieri*, tend to be the most frequently detected along with *A. restrictus* species, *Penicillium* (*P*. *aurantiogriseum*, *P. viridicatum*), and other closely related species [21,29]. Storage of maize has to be in well-constructed silos to avoid migrating moisture, which in turn limits aflatoxin production that occurs when the producing fungi, especially when *A. flavus*, is present. However, storage conditions in less economically developed countries are not the most suitable: in uninsulated metal silos, buildings with leaky roofs or dirt floors, or in wooden crates in the open air [26].

The economic and nutritional losses caused by the effects of contaminated grains are very diverse. They can affect germination capacity, alter natural coloration, cause bad odors, change chemical composition and reduce dry matter [30], cause heating, appearance of spots, alter flavor, and cause the production of mycotoxins [31,32,33,34]. Mycotoxins are considered an issue in this crop, as they are not fully removed when maize is subjected to industrial processes, even when the grain is exposed to high temperatures [35,36,37,38]. These contaminants are transmitted through the human and animal food chain via cereals to meat, milk and egg rations from animals exposed to these mycotoxins [39,40,41,42].

The presence of a toxin-producing fungus is not synonymous with the mandatory presence of the mycotoxin, since many factors in the substrate and environment determine its production. Similarly, the absence of any visible sign of a filamentous fungus does not guarantee that the grain is toxin-free, as the fungus may have been eliminated at some point in the process, but the toxin formed could still be present in the grain [43,44,45,46,47,48].

Maize is one of the most widely cultivated agricultural crops in the world, and in Ecuador it is widely used for animal husbandry and for feeding the local population. It is not easy to find studies that relate the development of fungi with the environmental conditions of the geographical regions and cultivated varieties of maize in Ecuador. Likewise, no studies have been found on the incidence and simultaneous incidence of multiple mycotoxins in a substrate. The aim of this study was to analyze the occurrence of fungi and mycotoxins in maize grown in different regions of Ecuador related to weather conditions such as temperature, relative humidity, precipitation, luminosity, among others.

## 2. Materials and Methods

### 2.1. Samples

INEN 1233:95 standard sampling methodology was used, with slight modifications, which were subject to the specific harvest and post-harvest practices of each sampled location [49].

The maize samples were taken in bulk from 50 producers and were collected from collection centers in localities of each region. The localities studied are representative of each region’s maize production. It comprises 29 localities distributed as 12 in the Andean region and 17 in the coast region (Figure 1).

The collection of samples from the 17 varieties of “duro” hard was carried out in different localities of the coast provinces with the highest production of hard maize in Ecuador: Guayas, Manabí, Los Ríos, and Loja. Moreover, 9 samples of the “suave” soft variety and 3 of the “morocho” were taken in the highlands provinces of Azuay, Cotopaxi, Bolívar, Chimborazo, Pichincha, Carchi, and Tungurahua (Table 1). In each of these locations, a sample was taken, and all analyses and records of each study variable were made.

### 2.2. Fungal Determination

Twenty-nine samples of maize from the coast and highlands of Ecuador were evaluated, and 24 grains belonging to each sample were disinfected in a 0.4% sodium hypochlorite solution for one minute, placed in Petri dishes containing Dicloran Glycerol 18% Agar (DG18), and incubated at 25 °C for 7 days for fungal detection. Analysis of multi-toxins was carried out using liquid chromatography coupled with mass spectrometry (LC-MS/MS). After the incubation period, the plates were examined, and the grains that showed fungal development were counted. The results were expressed as a percentage of internally infected grains, according to the methodology of [15]. The fungi grown from the samples were first isolated on plates containing Czapek Yeast Autolyzed Agar (CYA) to be identified through specific protocols for each genus.

### 2.3. Fungal Identification

*Aspergillus* sp. isolates were 3-point inoculated on the CYA and Malt Extract Agar (MEA) Petri dishes and incubated for 7 days at 25 °C. In parallel, CYA plates were inoculated in the same way, with the same isolates, and incubated at 37 °C for the same period. *Aspergillus* isolates belonging to the section *Aspergillus* (formerly *Eurotium*) were additionally cultured on Czapek Yeast Extract Agar and 20% Sucrose (CY20S) for 14 days at 25 °C.

Isolates of the genus *Penicillium* were 3-point inoculated on plates containing CYA, MEA, Yeast Sucrose Extract Agar (YESA), and Creatine Yeast Extract Agar (CREA) and incubated for 7 days at 25 °C. In parallel, CYA plates were inoculated with the same isolates and incubated at 5 and 37 °C for the same period.

After the incubation period, the fungi were identified according to the references indicated for each genus through the observation of macro and microscopic characteristics of the colonies (such as colony size, color of the verse and reverse side, texture, production of exudate and soluble pigment, shape, and ornamentation of microscopic structures, among others). Finally, the frequency of occurrence of each species was calculated based on the total number of fungi present in each sample.

Pitt and Hocking method [21] was used as a reference for *Aspergillus* and *Penicillium* identification, punctually complemented with other sources if necessary.

Considering the existence of morphological similarities among some species, an infrageneric identification until the section level was adopted in this manuscript, according to [50,51], which provide more information when compared to the genus level.

### 2.4. Mycotoxin Determination

The samples were coded and transported to the LAMIC Mycotoxin Analytic Laboratory of Universidad Federal de Santa Maria to detect and quantify mycotoxins. The mycotoxin detection was carried out by a mass spectrophotometer (API 5000 LC-MS/MS System), allowing access simultaneously to a total of 15 mycotoxins from the maize samples: 11 mostly associated with the genus *Fusarium* [Deoxynivalenol, 3-acetyl-deoxynivalenol and 15-acetyl-deoxynivalenol (DON, 3-Don and 15-DON), Zearalenone (ZEA), fumonisins B1 and B2 (FUMO B1, and FUMO B2), Nivalenol (NIV), Fusarone-x (Fusa-x), Diacetoxyscirpenol (DAS), T-2 and HT-2 toxins], and 4 associated to the genera *Aspergillus* [Aflatoxins B1, B2, G1 and G2 (AFB1, AFB2, AFG1, and AFG2)].

The analyses were performed using Sulyok et al.’s method (2006) adapted to laboratory conditions. Particularly an API 5000 LC-MS/MS System (Applied Biosystems, Foster City, CA, USA) equipped with a Turbo Ion Spray electrospray ionization (ESI) and 1200 Series HPLC System (Agilent, Waldbronn, Germany) were used. A total of 20 mL of acetonitrile:water solution (84:16, *v*/*v*) was added to a 5 g sample and conducted to a shaker (Lucadema, Sao Paulo, Brazil) for 1 h and 30 min at 70 rpm. The extract was then properly diluted (1:10), and 10 μL of the diluted extract was injected into a 1200 Series Infinity HPLC 1200 Series Infi (Agilent, Palo Alto, CA, USA) coupled with a 5500 QTRAP mass spectrometer (Applied Biosystems, Foster City, CA, USA) equipped with an electrospray ionization (ESI) source in positive mode. Chromatographic separation was performed at 30 °C using an Eclipse XDB-C8 column (4.6 × 150 mm, 5 μm particle diameter) (Agilent, Palo Alto, USA) with the mobile phase gradient composed of solutions of methanol: water:ammonium acetate (95:4:1, *v*/*v*/*v*) (solution A) and water:ammonium acetate (99:1, *v*/*v*/*v*) (solution B).

### 2.5. Data Analysis

Selection of explanatory variables was developed by correlation analysis (to determine redundant information).

The statistics used for the analysis of these variables were descriptive (mean, standard deviation, and skewness coefficient). For this purpose, the data obtained in each of the samples from all the localities under study were used. Additionally, the study was carried out by grouping the samples from the coast or Littoral and the Inter-Andean Valleys.

### 2.6. Quantitative Explanatory Variables (Numerical)

Table 2 shows the quantitative variables considered in the present study to determine the effect of environmental conditions on the fungi and the mycotoxins present in the maize kernels collected in the two regions studied.

### 2.7. Principal Component Analysis (PCA)

Principal Component Analysis (PCA) was used because the study presented many variables, and the aim was to reduce this number in order to explain the variability of the data. This statistical technique facilitates the interpretation of the results. The data set analyzed by Principal Component Analysis (PCA) includes 29 records of 10 variables: “Region”, “Precip”, “Grain_moisture”, “Aw (water activity)”, “Total_infection”, “*Fusarium*_X”, “MO (microorganisms)”, “*A. niger* (%)”, “temp” (average temperature), and “B1pB2”. The quantitative variable “B1pB2” was not considered for PCA determination, and the qualitative variable “Region” was considered in the analysis as illustrative.

Prior to performing the PCA, an outlier study of the variables considered was carried out using [52], ensuring that no data were outliers. Then the PCA model was generated by applying the algorithm included in the FactoMineR Package of the R 4.0.3 Statistical software [53]. For this, the variables were standardized to eliminate the influence of scales.

### 2.8. Regression Model

This model considers analyzing qualitative and quantitative data. To achieve the objective of the analysis, the R “step” function was used, this command provides an automated approach to select variables in the construction of regression models. The step function operates through two variants: forward and backward, described by [54].

In the forward approach, the model is initially built with one predictor variable, and the remaining variables are added one by one to improve the quality of the model according to the AIC (Akaike Information Criterion). The step function systematically evaluates all possible combinations of variables and selects the one with the lowest AIC index. This ensures optimal variable selection.

On the other hand, in the backward approach, the starting point is the complete model considering all the predictor variables (explanatory variables). The variables that contribute less to the quality of the model are eliminated one by one using the AIC criterion. This process is continuous until the final model is shown with the most relevant variables. The decision for the selection of one of the models to explain the variable of interest is the comparison between the AIC indexes of the models, the most adequate being the one with the lowest index.

In addition, to compare models based on AIC, it is essential to consider the coefficient of determination (R^2^) to assess the relative and absolute quality of the model. The R^2^ indicates the proportion of the variability in the presence of maize toxins that is attributed to the prediction variables included in the model. A higher R^2^ indicates a greater ability of the model to explain the observed variability.

The use of regression models is appropriate for this analysis because it allows examining the relation between predictor variables and the presence of toxins in maize in a quantitative manner. Regression models provide a mathematical representation of the relationship between variables. In addition, by using the “step” function of R, an optimal and efficient selection of variables is ensured, contributing to obtaining more accurate models that explain the presence of mycotoxins in maize.

Both methods (regression model and PCA) are complementary; regression provides direct assessment of the explanatory-predicted relationship, and PCA provides a global and exploratory view of the joint behavior of the variables. These two analysis methods facilitate interpretation prior to modelling and identify possible collinearities or redundancies that could negatively affect the quality and stability of the regression models.

### 2.9. Software Used

All statistical analyses and functions used in the analysis are part of the “stats” package of the statistical analysis software [55].

The software used for descriptive statistics, correlation analysis, and the creation of statistical models was R with its set of basic packages. In particular, the package stats was used to create the models [53].

## 3. Results and Discussion

### 3.1. Identification and Determination of Fungi

The results of the incidence of fungi and mycotoxins in the highland and coastal areas of Ecuador are shown in Figure 1. Only one maize sample of the total collected was free of fungi. In addition, in half (15/29) of the samples analyzed, all kernels were contaminated with fungi, with *Fusarium* and *Aspergillus* being the most frequently isolated fungal genera.

Several species of *Fusarium* have the characteristic of infecting crops before harvest, especially under favorable climatic conditions for the development of the fungus, such as rainy periods and high temperatures [56]. On the other hand, the genus *Aspergillus* is frequently associated with samples in storage, although some species of this fungus can also infect maize in the field [21].

In general, xerophilic *Aspergillus* section *Aspergillus* (syn. *Eurotium*) were predominant in all analyzed samples. However, *Fusarium* of the *Liseola* group (which includes *F. verticillioides*, *F. proliferatum*, and other related species) should be mentioned because they were isolated from all samples from the coast and only one sample from the highlands. In the samples from the coast, there was also a higher prevalence of the genus *Penicillium* compared to the highlands.

The relationship between fungi and maize has been extensively explained by the work of Pitt and Hocking [21], where it is pointed out that maize ears are enveloped in a strong protective husk that manages to reduce the attack of several fungi. However, the genus *Fusarium* is able to infect and invade the kernels. *F. graminearum* is a species that causes reddening of the kernels and husk, while *F. verticillioides* and *F. subglutinans* cause less aggressive infections and often occur as commensal fungi. No less important than diseases caused by *Fusarium* are infections caused by *A. flavus*, which can even spread into maize ears from the field. Depending on the size of the maize kernels, which can be quite large, drying occurs more slowly, allowing the establishment of fungal species in the pre- and postharvest stages. Indeed, xerophilic species of *Aspergillus* and *Penicillium* generally prevail together. Similarly, *F. verticillioides*, *F. semitectum*, and *F. proliferatum* are often persistent in stored maize kernels.

Species of toxigenic interest belonging to the genus *Aspergillus* and *Fusarium* are presented in Table 3.

### 3.2. Mycotoxins

The presence of mycotoxins was detected in 93.1% of the samples analyzed (27/29 units). Five classes of mycotoxins were determined with a predominance of fumonisins (68.9%, 20/29 samples with maximum levels of 6777 µg/kg), zearalenone (31%, 9/29 samples with maximum levels of 573 µg/kg), and aflatoxins (17.2%, 5/29 samples with maximum levels of 130 µg/kg). On the other hand, group B trichothecenes were present in 13.8% of the samples (4/29 units), while ochratoxin A and group A trichothecenes were not detected in any of the analyzed samples (Figure 2).

It should be noted that Miller [57] considered aflatoxins, ochratoxin A, fumonisins, trichothecenes (deoxynivalenol and nivalenol), and zearalenones as the most important mycotoxins worldwide. Most of these compounds were detected in the samples of this study. Since maize is considered a staple food in several countries of Meso and South America: Mexico, Peru, and Ecuador, once it is contaminated with mycotoxins, this implies the risk of diseases caused by the ingestion of this cereal [58].

The most prevalent mycotoxins in maize in Ecuador, according to this study, and which coincide with those most frequent in the United States and Nepal, are zearalenone, fumonisins, and aflatoxins (Table 4). Fusilier [59] reports these three mycotoxins as the most prevalent in samples from the state of Michigan. Similarly, Joshi [60] indicates that the highest incidence of mycotoxins, in a study in Nepal, corresponded to aflatoxins present in 76% of the samples analyzed, followed by fumonisins and zearalenone, present in 76% of all samples. These percentages are higher than those found in maize samples from Ecuador. According to [61], the limits for mycotoxins in maize (row grain) are total aflatoxins (B_1_ + B_2_ + G_1_ + G_2_) 15 µg/kg, fumonisin (B_1_ + B_2_) 4000 µg/kg, deoxinivalenol (DON) 2000 µg/kg for further processing and ochratoxin A 5 µg/kg. Currently, in Ecuador there is no regulation set for mycotoxin occurrence in maize, and the limits are variable in other South American countries [62].

Mallmann et al. [63] analyzed maize samples from the southern region of Brazil between 2011 and 2014 and determined fumonisin contamination in 25% of the samples, with an average of 6 µg/kg and a maximum of 24 µg/kg. These levels are below those recorded in Ecuador in the present study. On the other hand, in Brazil, refs. [62,63,64,65] recorded the incidence of fumonisins, aflatoxins, and zearalenones. Indeed, in products derived from maize collected in the Pernambuco state, characterized by a warm climate, fumonisins and aflatoxins were found in all samples, but zearalenone was not detected.

According to Yli-Mattila [66], in South Africa, considered one of the main maize producers on the African continent, grain produced by small-scale producers has a 100% fumonisin contamination rate, while the incidence of this mycotoxin in grain produced by large-scale producers is 98.6%. This confirms the relevance of this compound in maize.

On the other hand, B-type trichothecenes and zearalenone were the mycotoxins with the highest occurrence in temperate regions of northeastern China. In these regions, mycotoxin concentrations were highly variable from year to year, and this could be explained by major changes in precipitation or temperature during periods of maize crop susceptibility [67]. The most frequently detected group B trichothecene in the samples from Ecuador was nivalenol, but deoxynivalenol was not detected.

Yang et al. [68] also indicate that the most prevalent mycotoxin in maize samples in Taiwan was deoxynivalenol (group B trichothecene), followed by zearalenones, aflatoxins, and fumonisins. In addition, as in the present study, the samples were generally contaminated simultaneously by several mycotoxins.

### 3.3. Co-Occurrence of Several Mycotoxins

Up to three classes of mycotoxins were detected simultaneously in the same maize sample in Ecuador (Table 5). Fumonisins, trichothecenes B, and zearalenone were present in two of the samples analyzed. On the other hand, aflatoxins, fumonisins, and zearalenones were found in another sample. Mycotoxins were absent in the soft and “morocho” maize produced in the highlands, while on the coast, where hard corn is grown, the presence of several mycotoxins were identified (Table 5). This could have occurred due to the environmental conditions of the region, since the highlands are cold and with low relative humidity in comparison to the coast, where high temperature and relative humidity predominate and are favorable for fungal growth and mycotoxin synthesis [8,69].

Confirming the presence of multiple mycotoxins in food for human and animal consumption is of utmost importance, as they represent a health risk to the consumer. It should be noted that mycotoxins can have additive, synergistic, or antagonistic interactions. Joint occurrence of mycotoxins is frequent since toxigenic fungi are capable of synthesizing more than a single active compound, and likewise, several fungi can simultaneously infect maize kernels [15,67,68].

A global monitoring work indicated that 72% of grain samples were contaminated with more than one mycotoxin and demonstrated that the co-occurrence of these toxins is high worldwide [70]. In Europe, several studies describe that the joint incidence of mycotoxins is relatively high and that grains were found to be contaminated in a large percentage by trichothecenes, fumonisins, and zearalenone [70]. Another study revealed that 75–100% of the analyzed grain samples were contaminated by more than one mycotoxin and that this could affect the health of consumers in Europe [71].

Biscoto et al. [72] analyzed samples of maize kernels in Brazil and observed that co-occurrence was also very common, with 87% of the samples containing one or more mycotoxins at the same time. The mycotoxins with the highest recurrence were fumonisins, zearalenone and, slightly, to a lesser extent, aflatoxins.

Topi et al. [73] show in their studies on maize produced in Albania that two or more toxins were detected in all the samples evaluated, and the incidence was mainly of toxins associated with *Fusarium*. They also point out that the five samples contaminated with aflatoxins also had fumonisins and that the only sample they found with ochratoxin A also presented zearalenone.

The isolated or concomitant presence of aflatoxins and fumonisins is of particular concern for maize-based products because both are thermally stable mycotoxins and can therefore prevail after the fungi have been killed. This is a real challenge for safe food production [21].

### 3.4. Place of Origin and Fungal-Mycotoxin Interaction

A closer analysis of results presented in Table 5 shows significant differences in the incidence profile of fungi and mycotoxins in the two regions.

Due to its geographic position, Ecuador suffers little variation in temperature and sunshine throughout the year in the same region; however, the fluctuation in altitude, temperature, and humidity is wide between regions [8]. On the coast, the temperature ranges between 23 and 36 °C all year long: the rainy season is between December and May, while the dry season is between June and November. The highlands, on the other hand, have a rainy climate between November and April and a dry season from May to October. The temperature in this area varies between temperate and cold depending on the altitude. Thus, the incidence of fungi and mycotoxins is not only influenced by the variety of maize grown but also by the microclimates present in each region.

Regarding the fungi that have the potential to produce mycotoxins in maize, all the samples analyzed that had *Fusarium* section *Liseola* were also contaminated with fumonisins and vice versa.

Goertz et al. [74] ] report that *F. verticilloides* (section *Liseola*) is a species associated with maize in warmer and drier regions such as Spain or Italy. In terms of dispersal, the genus *Fusarium* was predominantly isolated from maize in Germany in 2006. It should be emphasized that high levels of fumonisins tend to occur when maize plants are subjected to water stress or suffer significant damage by insect pests [21,26].

In other areas, such as Africa, Dutton [75] describes the predominance of *F. verticilloides* in commercial maize throughout the continent, including South Africa, which is considered the major producer of fumonisins together with *F. proliferatum*. Yoshizawa [76] analyzed samples from Thailand and observed that most samples were contaminated with fumonisins produced by *Fusarium* sp. It should be noted that among the known fumonisin compounds, those of the B series (B1, B2, B3) are of most concern in relation to the incidence of toxicity. Fumonisin B1 is the most relevant since it has the highest prevalence and toxicity; it is also suspected to be carcinogenic to humans [77].

On the other hand, 44.4% (8/18) of the samples from the Ecuadorian coast that showed the presence of *Aspergillus* section *Flavi* had no aflatoxins. This *Aspergillus* section, as well as aflatoxins, was not detected in samples from the highlands. Aflatoxins were present in 27.8% (5/18) of the coastal samples, with varying levels between 1.68 and 132.3 µg/kg. *A. flavus* is considered the main source of aflatoxins, the most important mycotoxin worldwide in food production and in maize and its derivatives; it constitutes a problem of specific importance [15].

Magnoli et al. [78] found that maize samples from the Ecuadorian highlands’ region had higher infestations of *A. flavus*, and it was among the most prevalent fungi after *F. graminearum*, *F. verticillioides*, and *A. parasiticus*, fungi that were also described. These results disagree with those found in the present study, where potentially aflatoxin-producing fungi such as *Aspergillus* section *Flavi* were not detected in samples from the highlands, nor *Fusarium* section *Discolor*. Koletsi et al. [79], through the analysis of maize samples, observed that aflatoxins were present mainly in cereals produced in warm regions such as southern Europe, Africa, and south and southwest Asia.

The mycotoxin that predominated in the Ecuadorian highlands was zearalenone, present in 41.7% of the samples, while in the coastal region, fumonisins were present in 100% of the samples from that area and in only one sample from the highlands. It should be considered that the joint incidence was present in 22% of the samples from the coast, and this index was considerably lower in the highlands.

Surprisingly, no *Fusarium* section *Discolor* was detected in the samples from the highlands; however, the zearalenone-positive samples from this region were contaminated with *Fusarium* section *Arthrosporiella*. Moreover, the four zearalenone-positive samples were infected by *Fusarium* section *Discolor* and vice versa: only two of these four samples had *Fusarium* section *Arthrosporiella*.

Pitt and Hocking [21] report *F. graminearum* and *F. culmorum*, both belonging to section Discolor, as the main zearalenone-producing species, although some other isolates such as *F. equiseti* (section *Gibbosum*) and *Fusarium semitectum* (section *Arthrosporiella*) also possess this characteristic. This suggests that different species of Fusarium produce zearalenone in Ecuadorian maize: *Fusarium* section *Discolor* in the samples from the highlands and *Fusarium* section *Arthrosporiella* in the coastal region.

Weaver et al. [80] analyzed samples from the United States and established that 100% of the samples were infested with one or more mycotoxins, most of which were produced by fungi of the genus *Fusarium*, and especially represented was zearalenone.

Fumonisins and trichothecenes B were also detected in samples from the highlands in low frequency (7.69% in 1/13 samples).

In the samples from the coast, the most important mycotoxins were fumonisins, trichothecenes, and zearalenone. These mycotoxins come from fungal species that are highly prevalent in the samples, such as *Fusarium* section *Liseola* and *Arthroporiella*. In the southern region of Brazil, which includes the states of Rio Grande do Sul, Santa Catarina, and Paraná, which have a temperate climate combined with hot summers with the particularity of high humidity, rainfall, and relatively low temperatures are characteristics that favor the contamination of maize grains with fumonisins [81]. These results coincide with those found in the present study in the coastal region, where similar climatic conditions are found.

Ducos et al. [82] observed results similar to the present study when analyzing maize samples in Peru, where they visualized high levels of fumonisins associated with the presence of *Fusarium* spp. Although *Aspergillus niger* is a fungus that could infect maize and has the potential to produce fumonisins, the level of importance of this fungus in fumonisin contamination in this product is still unclear [21].

Studies by Fusilier et al. [59] indicate that the main mycotoxins prevalent in maize are deoxynivalenol, fumonisins B1 + B2, and zearalenone. This shows that mycotoxins produced by the same fungal species or species of the same genus are frequently found together in maize samples.

### 3.5. Principal Component Analysis (PCA)

This statistical technique was used to reduce the dimensionality of the original set of quantitative explanatory variables in order to identify underlying patterns, reveal relationships between variables, and highlight which variables contribute most to differentiating or grouping observations according to their variability.

### 3.6. Distribution of Variability (Inertia)

In order to determine the optimum number of Principal Components (dimensions) that explain the variability in the maize samples, the Elbow analysis was performed (Figure 3A), where it can be seen that the first four groupings/dimensions have a significant contribution to the variability. However, the first two dimensions (components) of the PCA explain 55.7% of the total variability. This percentage is relatively high, so dimensions 1 and 2 are considered the most relevant in this study.

Considering the 11 variables studied in the 29 localities belonging to the coast and the highlands, the grouping of maize types according to production region can be clearly observed. Quadrants 2 and 3 are dominated by maize samples collected in the Sierra (highlands: red), while quadrants 1 and 4 are dominated by samples from the Costa (coast) (Figure 3B). In the highlands, groups of very similar maize samples (3, 7) and others very different (5, 2, and 11) are observed. In the coast, maize samples from localities 15 and 17 are very similar, while those from localities 13, 24, and 26 had a very different behavior.

In Figure 3B,C, individuals 13, 17, 22, and 19 (quadrant 4) are characterized by high positive values in component 1, which contrast with individuals 5, 8, 7, 3, and 1 (quadrant 3). The group formed by the observations of localities 13, 17, 22, and 19, is characterized by sharing high values for the variables temp_, Fusarium_X, Aw, Grain_moisture, Total_infection, and B1pB2. While the group of observations from localities 5, 8, 7, 3, and 1 share low values for the variables Fusarium_X, Total_infection, temp_, Grain_moisture, Aw, and B1pB2. It is important to highlight that the region factor is highly correlated with the water activity parameter in dimension 1.

Based on the number of groupings (4) shown in Figure 3D, it can be indicated that group 1 is formed by the maize samples from locations 1, 3, 5, 7, and 8. This group is characterized by component 1, which has negative values for all variables, and with respect to component 2, it has negative values for the variables total_infection, temp_, and MO, and positive values for Grain_moisture, AW, FusariumX, and B1pB2. Group 2 is only formed by the sample from locality 2. Group 3 is composed of the maize samples obtained at locations 11, 24, 26, 27, and 28, which are characterized by high values for the variables MO and A. niger (%) and low values for water activity (Aw). Finally, group 4, composed of maize samples from locations 13, 15, 16, 17, 19, and 22, is characterized mainly by high values for the variables: Grain_moisture, temp_, Aw, Fusarium_X, B1pB2, and Total_infection.

Developing ears of maize are encased in a strong, protective husk, which reduces invasion by fungi. *Fusarium* is the principal pathogenic fungal genus causing spoilage of the ear in maize, the most commonly occurring species being *F. graminearum, F. verticillioides* (=*F. moniliforme*), and *F. subglutinans* [21].

Of no less importance than the *Fusarium* diseases is the fact that the mycotoxigenic fungus *Aspergillus flavus* also invades maize, although it is not considered to be a true pathogen. In the early literature, *A. flavus* was regarded only as a storage fungus, but by 1970 the realization came that freshly harvested maize in the southeastern United States was sometimes infected by *A. flavus* with the consequent production of aflatoxins [21].

Maize cobs and kernels are relatively large. Moist conditions at harvest often result in slow drying. In consequence, both preharvest and postharvest fungi may become well established. Lichtwardt and Barron Refs. [83,84] carried out a very thorough study of the mycoflora of dried and stored maize in Iowa. Lichtwardt [83] identified the internal flora of surface-disinfected maize grains both in sterile moist chambers and on malt salt agar (6% NaCl). In addition, samples were ground and dilution plated. Approximately 50 genera were recognized. A combination of isolation methods enabled Barron [84] to estimate the relative importance of the isolated genera in the spoilage of stored maize. *Eurotium* species, especially *E. rubrum*, *E. amstelodami*, and *E. chevalieri*, were the most significant, together with *Aspergillus restrictus* and *Penicillium* spp., especially *P. aurantiogriseum* and *P. viridicatum* and closely related species. In maize samples from Thailand, the most encountered storage fungi were again *Eurotium* species (*E. chevalieri*, *E. rubrum*, and *E. amstelodami*) but *Wallemia sebi*, *A. flavus*, *A. wentii*, *A. tamarii*, and *A. niger* were also present in a significant number of samples [21].

*Penicillium* species occurring in maize both preharvest and in storage, and the factors that influenced their role as spoilage fungi, were investigated by [85]. Some common preharvest species, such as *P. funiculosum*, were rarely isolated later; species such as *P. citrinum* and *P. oxalicum* were always commonly present. Others again, such as *P. aurantiogriseum* and *P. viridicatum*, were almost exclusively associated with the stored grain.

Fungi acquired in the field, particularly *F. verticillioides*, *F. proliferatum*, *F. oxysporum*, and *A. flavus*, can persist in maize, and the molds, and their toxins may be carried through to maize products such as flour, grits, maize chips, tortillas, breakfast cereals, etc. [21]. In Central America, the process of nixtamalization is commonly used in preparation of meals based on maize. Nixtamalization is a centuries old process in which maize is soaked and then cooked with ash or lime high in alkali. It removes almost all fumonisins (and aflatoxins), resulting in tortillas and other maize-based foods being substantially free of these mycotoxins [86].

### 3.7. Selection of Explanatory Variables by Correlation Analysis to Determine Redundant Information

The influence of the explanatory (independent) variables on the amount of mycotoxins present in maize kernels harvested in the different locations was determined through the degree of linear association between the quantitative explanatory variables. The correlation coefficients (>0.5) between all quantitative explanatory variables are shown in Table 6.

The most significant correlation coefficients are presented in Table 7. The information presented in Table 6 shows that there is a potential overlap of information between the percentage of grain moisture, the amount of Fusarium_X, and the average temperature of the collection site. There is evidence that environmental factors such as temperature, humidity, and precipitation, aong others (intensity and distribution), influence the presence of fungi in cereals such as maize, which produce mycotoxins, affecting food safety [87,88], and people’s and animal’s health if consumed [89].

In all the cases indicated in the table, a significance test was applied to the value of the correlation coefficient, and in each case the value was found to be statistically non-zero with 95% confidence. Subsequently, the stepwise multiple regression algorithm was applied, in its two possible variants or directions, thus considering the Akaike information criterion (AIC) as a comparison parameter and obtaining the statistical models.

### 3.8. Linear Regression Models: Forward and Backward

The purpose of applying this statistical analysis technique in the study is to model and quantify how one or more explanatory variables (quantitative and qualitative) directly influence a specific variable of interest, in this case the amount of mycotoxins, allowing for clear predictions and estimates of specific relationships.

The Stepwise linear regression method used to determine the variables that have the greatest impact on the presence of mycotoxins in maize is that described by [54].

#### 3.8.1. Aflatoxin B1 + B2 + G1 + G2

The variable that has the greatest significance in the presence of aflatoxin B1 + B2 + G1 + G2 in maize kernels is grain moisture. Both forward (inclusion) and backward (exclusion) algorithm models incorporate the information of this variable (Table 7). Both models have significance to explain the behavior of aflatoxin B1 + B2 + G1 + G2 in the maize sample. However, the exclusion (backward) model explains approximately 50.6% more of the variability of this mycotoxin and has a higher relative quality since the AIC (117.2318) is lower than the forward model (215.5036).

#### 3.8.2. Fumonisin B1 + B2

The variable Fumonisin B1 + B2, in principle, has a significant linear relationship with the variables Humedad_grano, Aspergillus Section flavi, Fusarium Section liseola, and temp_. However, only the forward model revealed significant individual coefficients for moisture grain and Aspergillus Section flavi (Table 7). The backward model, despite including all four variables, did not reveal significance in its individual coefficients. With these results, we can infer that, although the backward model increases the explained variability and relative quality, such increases do not justify losing the simplicity of the model; therefore, the best model is the forward model.

#### 3.8.3. Ochratoxin A (OTA)

For ochratoxin A (OTA), the backward model is the most efficient, since it has the highest coefficient of determination (R2) and the lowest Akaike index (104.4055) when compared to the forward model (Table 7). In addition, the backward model includes the largest number of variables (environmental and microbiological) to explain the presence of OTA in maize kernels. This increases the percentage of variability explained by the full set of regressor variables and significantly improves the relative quality of the model.

#### 3.8.4. Trichothecenes B**

The qualitative explanatory variables with the strongest linear relationship with trichothecenes B are Dematiaceus and Fusarium section Discolor; both appear as part of the regressor variables in the inclusion and exclusion models, with significant coefficients. As for the models, both have identical characteristics in terms of explained variability and relative quality, and their sets of regressor variables have slight modifications. Therefore, either model is useful to explain the presence of trichothecenes B in maize kernels. If one of the models must be decided, the inclusion model (forward model) would be the most appropriate since it considers fewer variables with non-significant individual coefficients.

#### 3.8.5. Zearelenone (ZEA)

The presence of the mycotoxin Zearelenone (ZEA) in maize kernels is better explained with the backward model, since it slightly increases the variability explained (Table 8). The relative quality with respect to the forward model is superior and considers two variables (OM and temp_) with non-significant individual coefficients.

## 4. Conclusions

Mycological and mycotoxicological analysis of maize samples from the Ecuadorian Sierra revealed a wide range of fungal contamination (93.3% of the samples) and mycotoxin contamination (90% of the samples), showing the existence of different contamination profiles depending on the origin of the sample. The genera *Fusarium* and *Aspergillus*, which include species with mycotoxin-producing potential, were the most frequent. Four classes of mycotoxins were detected with the following representativeness in the samples: fumonisins (66.6%), zearalenone (30%), aflatoxins (16%), and group B trichothecenes (13.3%).

Contamination by different fungi and joint incidence of several mycotoxins were observed in some samples. The joint incidence of different types of mycotoxins was more frequent in samples from the coast: aflatoxins and fumonisins were the most found in parallel. Up to three types of mycotoxins were detected coexisting in the same sample.

The regression models indicate that the presence of mycotoxins in maize kernels is mainly influenced by grain moisture, temperature, and the presence of *Aspergillus* and *Fusarium* fungi. In general, it is possible to infer that the backward model is better than the forward model, since it allows a more accurate prediction of the presence of mycotoxins, as it has a lower Akaike index than the forward model.

This important and unpublished information on the microbiological and mycotoxigenic quality of Ecuadorian maize is of great relevance since it could represent a potential danger to the consumer due to the high consumption of maize by the local population. Indeed, these compounds are associated with toxic and carcinogenic effects.

This work aims to promote discussion on the contamination of maize kernels by fungi and mycotoxins present in an important way in a staple food for the population studied. In the same way, information from other studies relevant to this topic can be considered as a reference for future research that contributes to the control of fungi infecting maize and food safety for the Ecuadorian population.

## Figures and Tables

**Figure 1 foods-14-02630-f001:**
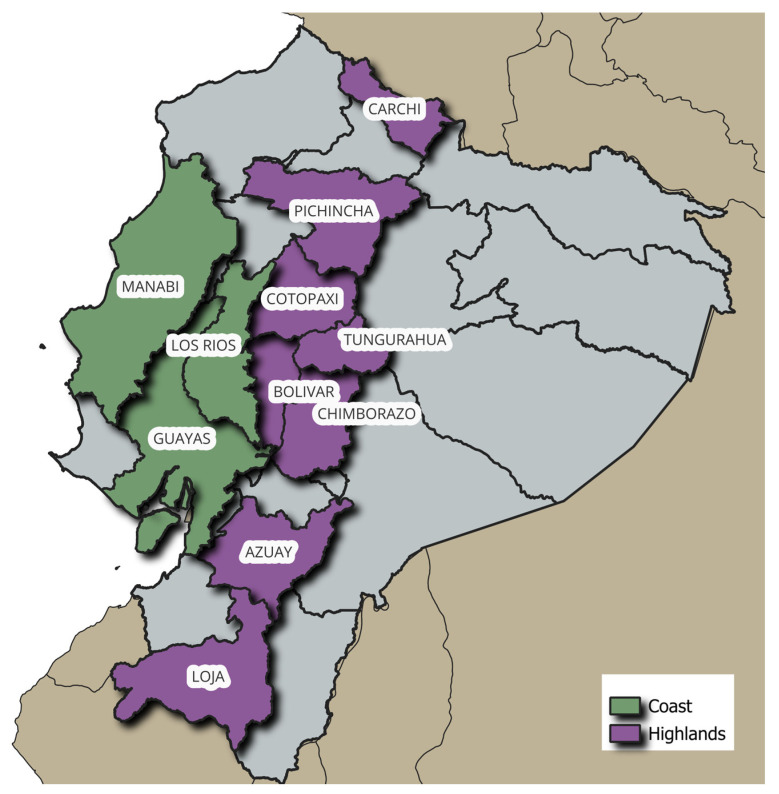
Samples collected in the Ecuadorian regions.

**Figure 2 foods-14-02630-f002:**
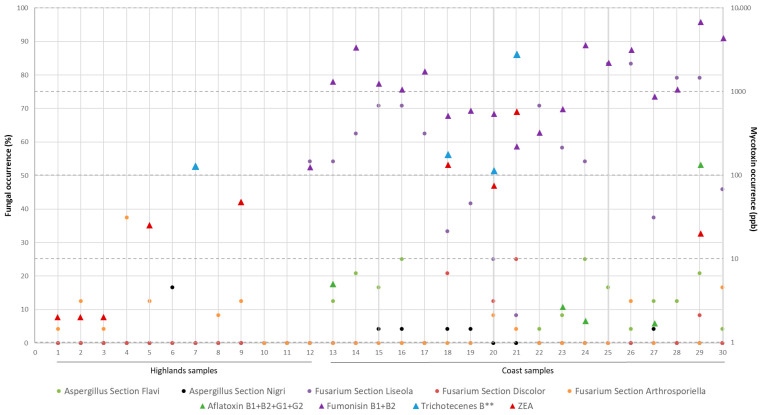
Incidence of groups of potentially toxigenic fungi and mycotoxins in maize samples collected in the coast and highlands of Ecuador.

**Figure 3 foods-14-02630-f003:**
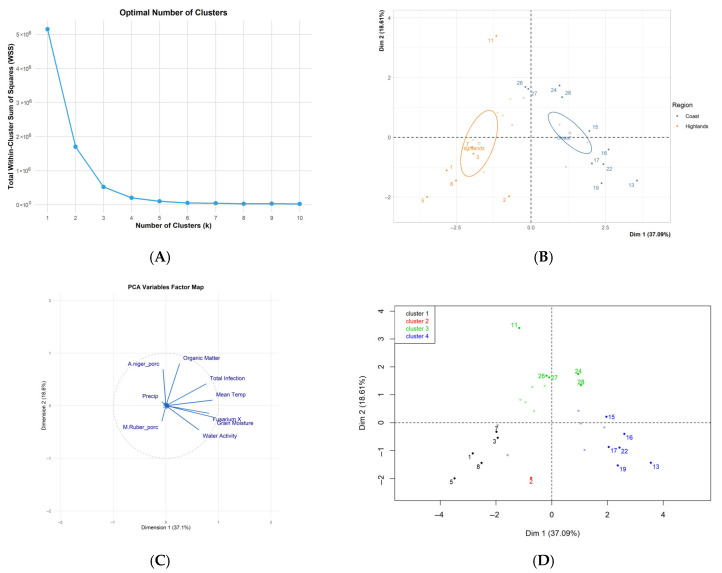
Principal Component Analysis. (**A**): elbow analysis indicating the dimensions that explain the variability of the maize samples studied. (**B**): map of maize origin as a function of collection region (squares are centroids/regions), the bold dots represent a higher incidence in explaining variability, and the unlabeled vectors correspond to qualitative variables that were not considered in the construction of the linear combinations. (**C**): black-colored variables are those considered in the PCA; the labeled ones are those of greater contribution in the determination of the projection plane. (**D**): hierarchical grouping of the maize studied according to the variables analyzed. The color gradient corresponds to those that have the greatest incidence in explaining variability.

**Table 1 foods-14-02630-t001:** Description of the factors and study levels of the maize kernels considered in the mycotoxin study.

Factors Under Study	Factors Description	Factor Levels	Observation (Number)	Sample (%)
* Region	Region of the country where the sample was collected.	Coast 1	17	58.6
		Highlands 2	12	41.4
* Grain type	Type of grain starch	Soft	17	586
		Semi-hard (Morocho)	3	10.3
		Hard	9	31.0
* Type of drying	Grain-drying process	Artificial	12	41.4
		Plant	3	10.3
		Sun	14	48.3
* Moisture	Amount of water in the grain	High (>15%)	8	27.6
		Low (<15%)	21	72.4
* Postharvest grain treatment	Pesticide	Without	27	93.1
		With	2	6.9

1-coast provinces: Guayas, Manabí, Los Ríos, and Loja. 2-highlands provinces of Azuay, Cotopaxi, Bolívar, Chimborazo, Pichincha, Carchi, and Tungurahua. * Qualitative variables.

**Table 2 foods-14-02630-t002:** Quantitative variables evaluated for the Ecuadorian maize study.

Characteristics of the Site and of the Maize Grain	Description
Height (m)	Altitude of the collection site, in relation to sea level.
Precipitation (mm/year)	Pluvial precipitation at the collection site.
Average temperature (°C)	Average temperature of the collection site.
Grain moisture (%)	Grain moisture after drying.
Water activity (Aw)	Grain water activity.
Total infection (%)	Total percentage of fungal infection.
Fusa-X	Mycotoxin produced by *Fusarium* fungus
Microorganisms	Counting of microorganisms present in maize grain

**Table 3 foods-14-02630-t003:** Species belonging to the genera of *Aspergillus* and *Fusarium* according to the series and sections to which they belong.

Genus	Infragenus	Species by Morphology
*Aspergillus*	Series *Flavi*	*A. flavus*, *A. parasiticus*, *A. arachidicola*, *A. minisclerotigenes*, *A. pipericola*, *A. novoparasiticus*, *A. austwickii*, *A. aflatoxiformans*, *A. transmontanensis*, *A. sergii*, *A. krugerii*, *A. mottae*, and *A. subflavus.*
	Series *Nigri*	*A. neoniger*, *A. costaricensis*, *A. vadensis*, *A. eucalypticola*, *A. pulverulentus* (=*A. tubingensis*), *A. tubingensis*, *A. luchuensis*, *A. piperis*, *A. lacticoffeatus* (=*A. niger*), *A. niger*, and *A. welwitschiae*
*Fusarium*	Section *Arthrosporiella*	*F. camptoceras*, *F. semitectum*, *F. avenaceum*, *F. chlamidosporum*, *F. pallidoroseum*, *F.*, *F. pallidoroseum*, *F. polyphialidicum*, *F. chlamidosporum*, *F. pallidoroseum*, and *F. polyphialidicum*
	Section *Discolor*	*F. heterosporum*, *F. reticulatum*, *F. sambucinum*, *F. culmorum*, *F. graminearum*, and *F. cerealis* (=*F. crookwellense*)
	Section *Liseola*	*F. verticillioides* (=*F. moniliforme*), *F. proliferatum*, *F. subglutinans*, *F. anthophilum*, *F. napiforme*, *F. succisae*, *F. dlamini*, *F. beomiform*, and *F. annulatum.*

Note: Relevant species for mycotoxin production are underlined.

**Table 4 foods-14-02630-t004:** Occurrence of different mycotoxins in samples of maize from the highlands and coast of Ecuador.

	Highlands (n = 12)			Coast (n = 18)			Total (n = 30)		
Mycotoxins	Occurrence (%)	Mean ± SD (µg/Kg)	Range (µg/Kg)	Occurrence (%)	Mean ± SD (µg/Kg)	Range (µg/Kg)	Occurrence (%)	Mean ± SD (µg/Kg)	Range (µg/Kg)
Aflatoxins (B1 + B2 + G1 + G2)	0	-	-	27.8	7.97 ± 31.1	1.70–132	16.7	4.78 ± 24.1	1.70–132
AFB1	0	-	-	27.8	7.67 ± 29.9	1.7–127	16.7	4.60 ± 23.1	1.70–127
AFB2	0	-	-	5.6	0.30 ± 1.24	5.30–5.30	3.3	0.18 ± 0.96	5.30–5.3
AFG1	0	-	-	0	-	-	0.0	-	-
AFG2	0	-	-	0	-	-	0.0	-	-
Fumonisins (B1 + B2)	8.3	10.4 ± 36.1	125–125	100	1862 ± 1750	221–6770	63.3	1121 ± 1627	125–6770
FB1	8.3	10.4 ± 36.1	125–125	100	1431 ± 1350	176–5280	63.3	862 ± 1253	125–5280
FB2	0	-	-	100	431 ± 409	45.4–1490	43.3	258.9 ± 380	45.4–1490
Trichothecenes A	0	-	-	0	-	-	0	-	-
DAS	0	-	-	0	-	-	0	-	-
HT-2	0	-	-	0	-	-	0	-	-
T2	0	-	-	0	-	-	0	-	-
Trichothecenes B	8.3	10.6 ± 36.6	127–127	16.7	170 ± 649	113–2764	13.3	106 ± 504	113–2764
15-DON	0	-	-	0	-	-	0	-	-
3-DON	0	-	-	0	-	-	0	-	-
DON	0	-	-	0	-	-	0	-	-
Fusa-x	0	-	-	5.6	14.1 ± 59.9	254–254	3.3	8.47 ± 46.4	254–254
NIV	8.3	10.6 ± 36.6	127–127	16.7	155 ± 589	113–2510	13.3	97.5 ± 458	113–2510
Zearalenone	41.7	6.55 ± 14.7	2–47.4	22.2	44.5 ± 136	20.0–573	30.0	29.3 ± 106	2.00–573

AF = aflatoxin; F = fumonisin; DAS = Diacetoxicirpenol; HT-2 = HT-2 toxin; T2 = T2-toxin; DON = Deoxinivalenol; Fusa-x = Fusarone x; NIV = nivalenol. Analytical limits: AF LOQ = 1.0 µg/Kg for each one; LOD: AFB1 = 0.42 µg/Kg; AF B2 = 0.59 µg/Kg; AFG1 = 0.64 µg/Kg and AFG2 = 0.58 µg/Kg; FB1: LOQ = 125 µg/Kg; LOD = 10 µg/Kg; FB2: LOQ = 125 µg/Kg; LOD = 20 µg/Kg; DON: LOQ = 200 µg/Kg; LOD = 50 µg/Kg; TRIC (all): LOQ = 100 µg/Kg; LOD = 60 µg/Kg. Range = range of contamination among positive samples; = below the Limit of Detection (LOD).

**Table 5 foods-14-02630-t005:** Natural co-contaminating groups of mycotoxins in maize samples from the highlands and coast of Ecuador.

Co-Occurring Toxins	Highlands (n = 12)	(%)	Coast (n = 18)	(%)	Total (n = 30)	(%)
Aflatoxins + Fumonisins	0	0	4	22	4	13
Aflatoxins + Fumonisins + Zearalenone	0	0	1	6	1	3
Fumonisins + Nivalenol + Zearalenone	0	0	2	11	2	7
Fumonisins + Fusarone x + Nivalenol + Zearalenone	0	0	1	6	1	3

**Table 6 foods-14-02630-t006:** Significant correlation coefficients (>0.5) between descriptor variables (dependent variables).

Comparative Variables	Correlation Coefficient
Grain moisture vs. kernels infected with fungi (%)	0.67
Grain moisture vs. *Aspergillus* Section *Flavi*	0.53
Grain moisture vs *Fusarium* Section *Liseola*	0.63
Grain moisture vs. *Penicillium* sp.	0.55
Humidity_grain vs. Temperature	0.72
Kernels infected with fungi vs. *Aspergillus* Section *Flavi*	0.53
Kernels infected with fungi vs. *Fusarium* Section *Liseola*	0.64
Kernels infected with fungi vs. Temperature	0.72
*Aspergillus* Section *Flavi* vs. *Fusarium* Section *Liseola*	0.65
*Aspergillus* Section *Flavi* vs. Temperature	0.78
*Aspergillus wentii* vs. *Rhizopus* sp.	0.55
*Dematiaceus* vs. *Fusarium* section *Discolor*	0.73
*Fusarium* Section *Liseola* vs. Temperature	0.88

Note: Coefficients whose absolute value exceeds 0.5 have been considered as significant. In all the cases indicated in the table, a significance test was applied to the value of the correlation coefficient, and in each case the value was found to be statistically correlated with 95% confidence.

**Table 7 foods-14-02630-t007:** Correlation coefficients between explanatory variables.

Comparative Variables	Correlation Coefficient
Humidity_grain vs. Aw	0.65
Moisture_grain vs. *Fusarium*_X	0.70
Humidity_grain vs. temp_average	0.75
Total_infection vs. temp_average	0.66
*Fusarium*_X vs. temp_average	0.70

Note: Coefficients whose absolute value exceeds 0.6 have been considered as significant.

**Table 8 foods-14-02630-t008:** Variables considered in the linear regression models to explain the presence of the mycotoxins studied in different types of maize in two regions of Ecuador.

Toxins	Model	Significant Variables Explaining Mycotoxin Behavior	Coefficient of Determination (Adjusted R^2^)	Akaike Index (AIC)
Aflatoxin B1 + B2 + G1 + G2	Forward	‘Humidity_grain’ + ‘Region’	0.4751	215.5036
	Backward	Drying + Grain_type + Crop_moisture + Treated + Precip + Grain_moisture + ‘Kernels infected with fungi’ + ‘MO’ + ‘*Aspergillus* Section *Aspergillus*’ + ‘*Aspergillus* Section *Flavi*’ + ‘*Aspergillus* Section *Nigri*’ + ‘*Aspergillus wentii*’ + ‘*Cladosporium* sp.’+ ‘*Dematiaceus*’ + ‘*Fusarium* Section *Liseola*’ + ‘*Fusarium* Section *Discolor*’ + ‘*Penicillium* sp.’ + ‘*Rhizopus* sp.’+ temp_’	0.9812	117.2318
Fumonisin B1 + B2	Forward	Humedad_grano + ‘*Rhizopus* sp.’ + ‘*Aspergillus* Section *Flavi*’ + ‘*Aspergillus* Section *Nigri*’ + ‘*Aspergillus wentii*’ + Precip	0.7724	395.3998
	Backward	Drying + Grain_type + Crop_moisture + Treated + Precip + Grain_moisture + MO + ‘*Aspergillus* Section *Flavi*’ + ‘*Aspergillus* Section *Nigri*’ + ‘*Aspergillus wentii*’ + *Dematiaceus* + ‘*Fusarium* Section *Liseola*’ + ‘*Fusarium* Section *Discolor*’ + ‘ *Penicillium* sp.’ + ‘*Rhizopus* sp.’ + temp_	0.8311	382.872
Trichothecenes B	Forward	*Dematiaceus* + MO + ‘Kernels infected with fungi (%)’ + ‘*Rhizopus* sp.’ + Drying + Region + ‘*Fusarium* Section *Discolor*’ + Humedad_grano + ‘*Cladosporium* sp.’ + Treated + ‘*Penicillium* sp.’ + Crop_moisture + Grain_type + ‘*Fusarium* Section *Arthrosporiella*’ + temp_+ ‘*Aspergillus* Section *Nigri*’ + ‘*Fusarium* Section *Liseola*’ + ‘*Aspergillus wentii*’ + Precip + ‘*Aspergillus* Section *Aspergillus*’ + ‘*Aspergillus* Section *Flavi*’	1	86.49337
	Bakward	Drying + Grain_type + Crop_moisture + Treated + Precip Grain_moisture + ‘Kernels infected with fungi (%) + MO + ‘*Aspergillus* Section *Aspergillus*’ + ‘*Aspergillus* Section *Flavi*’ + ‘*Aspergillus* Section *Nigri*’ + ‘*Aspergillus wentii*’ + ‘*Cladosporium* sp.’ + *Dematiaceus* + ‘*Fusarium* Section *Liseola*’ + ‘*Fusarium* Section *Discolor*’ + ‘*Fusarium* Section *Arthrosporiella*’ + ‘*Penicillium* sp.’+ ‘ *Rhizopus* sp.’ + temp_	1	86.49337
Zearelenone (ZEA)	Forward	*Dematiaceus* + ‘*Fusarium* Section *Discolor*’ + Drying + Region + ‘*Aspergillus* Section *Aspergillus*’ + Treated + ‘*Penicillium* sp.’ + MO	0.996	174.0771
	Backward	Drying + Grain_type + Crop_moisture + Treated + Precip + Grain_moisture + ‘Kernels infected with fungi (%) + MO + ‘*Aspergillus* Section *Aspergillus*’ + ‘*Aspergillus* Section *Flavi*’ + ‘*Aspergillus* Section *Nigri*’ + ‘*Aspergillus wentii*’ + ‘*Cladosporium* sp.’ + *Dematiaceus* + ‘*Fusarium* Section *Discolor*’ + ‘*Penicillium* sp.’ + ‘*Rhizopus* sp.’ + temp_	0.9999	67.8733

## Data Availability

The original contributions presented in this study are included in the article. Further inquiries can be directed to the corresponding author.

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
