# Peer review of "Toxigenic Fungi and Co-Occurring Mycotoxins in Maize (*Zea mayz* L.) Samples from the Highlands and Coast of Ecuador"

_foods, 2025, doi:10.3390/foods14152630_

Round 1
Reviewer 1 Report
Comments and Suggestions for Authors
The manuscript titled "Toxigenic Fungi and Co-Occurring Mycotoxins in Maize (Zea mays L.) Samples from the Highlands and Coast of Ecuador" presents a comprehensive mycological and mycotoxicological investigation of maize collected from 29 localities across two major regions of Ecuador—the Coast and the Highlands. Using morphological identification and LC-MS/MS analysis, the study reveals a high prevalence of fungal contamination (93.3%) and mycotoxin occurrence (90%), with fumonisins, zearalenone, and aflatoxins being the most frequently detected. Statistical analyses including Principal Component Analysis (PCA) and linear regression were applied to identify environmental and postharvest variables influencing contamination levels. The authors emphasize that regional differences, especially in moisture content, temperature, and fungal profile, significantly affected the presence and co-occurrence of mycotoxins.
Comments
- The discussion should address how the detected levels of mycotoxins (particularly aflatoxins, fumonisins, and zearalenone) compare with established regulatory limits (e.g., Codex Alimentarius, European Union, or Ecuadorian standards). This would enhance the interpretation of the public health implications.
- The manuscript should report on the analytical performance of the LC-MS/MS method, including the limits of detection (LOD) and quantification (LOQ) for each mycotoxin analyzed.
- The mention of Table 5 in lines 551–552 should be reviewed. Table 5 presents information on co-occurring mycotoxins, which may not be directly relevant to the context of that sentence.
- The linear regression models used to examine the relationships between environmental/microbial variables and mycotoxin presence are valuable. However, it is recommended that the authors include scatter plots showing observed versus predicted values for aflatoxins (B1+B2+G1+G2), fumonisins (B1+B2), OTA, trichothecenes B, and zearalenone to support model validation and visual assessment of predictive performance.
- The study reports high levels of both aflatoxins and fumonisins in coastal maize samples, alongside frequent isolation of Aspergillus flavus and Fusarium spp. The discussion would benefit from elaborating on the documented interactions between these species, as previous research has shown that co-infection can lead to enhanced production of both toxin types.
Author Response
Comment 1. Done, it was included in the manuscript
Comment 2. Done, it was included in the manuscript
Comment 3. We apologize for this mistake, the correct table was refered.
Comment 4. We appreciate the suggestion, this would be relevant if we were seeking to validate the predictive methodology. However, in our study the objective was explanatory type, oriented to analyze the relationships between environmental/microbiotic variables and the presence of mycotoxins. For this reason, we chose to present the coefficients of determination (R²) and the Akaike Information Criterion (AIC) values, these allow to evaluate the absolute and relative quality of the models. We consider that this information is sufficient for the purposes of this work.
Comment 5. The suggestion was added in the document.
All the changes are in red in the manuscript
Reviewer 2 Report
Comments and Suggestions for Authors
I have read with great interest your comprehensive investigation of fungal species and mycotoxin profiles in maize harvested from various regions of Ecuador. The effort to analyze the interactions among environmental, physiological, and varietal factors is particularly commendable.
I think the research is very interesting, but I have a few minor suggestions:
-
Inter‐varietal Analysis
In Lines 126–131, you describe sampling three principal maize types (duro, suave, and morocho), and in the Introduction (Lines 50–60) you detail their differing cultivation practices and post‐harvest characteristics. However, subsequent analyses of differences between these varieties appear somewhat limited. Since maize variety inherently reflects geographic and physiological variables, it may be worthwhile to explore whether mycotoxin distribution correlates strongly with cultivar choice. -
Food‐Safety Recommendations
Should a strong correlation be confirmed, I suggest enhancing the Discussion with direct guidance on food‐safety monitoring. For instance, suave maize is commonly milled into cornmeal and further processed into tortillas and related products. You might therefore recommend which downstream products warrant heightened scrutiny for specific mycotoxin classes based on your findings. -
Validation via Commercial Samples
Finally, if resources permit, validating your survey results by sampling and testing domestically marketed maize products would lend additional real‐world relevance and robustness to your conclusions.
Thank you for your valuable contribution to the field. I hope these comments prove helpful as you refine your manuscript.
Author Response
Comment 1. Your suggestion was included in the manuscript
Comment 2. Your suggestion was included in the manuscript, considering the Codex Alimentarius.
Comment 3. Unfortunately, we cannot proceed with these additional analyses. This is important for future research.
All changes are in red in the manuscript.
Round 2
Reviewer 1 Report
Comments and Suggestions for Authors
Most comments were addressed successfully by the author.